# Survival of *Lactobacillus*
*salivarius* CECT 4063 and Stability of Antioxidant Compounds in Dried Apple Snacks as Affected by the Water Activity, the Addition of Trehalose and High Pressure Homogenization

**DOI:** 10.3390/microorganisms8081095

**Published:** 2020-07-22

**Authors:** Cristina Gabriela Burca-Busaga, Noelia Betoret, Lucía Seguí, Ester Betoret, Cristina Barrera

**Affiliations:** 1Instituto Universitario de Ingeniería de Alimentos para el Desarrollo, Universitat Politècnica de València, 46022 Valencia, Spain; cribur@posgrado.upv.es (C.G.B.-B.); noebeval@tal.upv.es (N.B.); lusegil@upvnet.upv.es (L.S.); 2Instituto de Agroquímica y Tecnología de Alimentos, Consejo Superior de Investigaciones Científicas, 46980 Paterna, Spain; ester.betoret@csic.es

**Keywords:** *Lactobacillus**salivarius* spp. *salivarius*, water activity, trehalose, high pressure homogenization, antioxidants, hot air drying, freeze-drying

## Abstract

Survival of probiotic microorganisms in dried foods is optimal for water activity (*a*_w_) values between 0.1 and 0.3. Encapsulating and adding low-molecular weight additives can enhance probiotic viability in intermediate *a*_w_ food products, but the effectiveness of sub-lethal homogenization is still not proven. This study evaluates the effect of 10% (*w/w*) trehalose addition and/or 100 MPa homogenization on *Lactobacillus*
*salivarius* CECT 4063 counts and antioxidant properties of apple slices dried to different water activity values (freeze-drying to a *a*_w_ of 0.25 and air-drying at 40 °C to a *a*_w_ of 0.35 and 0.45) during four-week storage. Optical and mechanical properties of dried samples were also analyzed. Freeze-drying had the least effect on the microbial counts and air drying at 40 °C to a *a*_w_ of 0.35 had the greatest effect. Antioxidant properties improved with drying, especially with convective drying. Decreases in both microbial and antioxidant content during storage were favored in samples with higher water activity values. Adding trehalose improved cell survival during storage in samples with a water activity of 0.35, but 100 MPa homogenization increased the loss of viability in all cases. Air-dried samples became more translucent and reddish, rather rubbery and less crispy than freeze-dried ones.

## 1. Introduction

Functional foods are those that, beyond their nutritional value, benefit human health by improving it, and even reducing the incidence of certain diseases. This includes everything from fruits and vegetables, with a naturally high content of antioxidant compounds [1], to design foods, in which one or more ingredients have been added, removed, concentrated or diluted.

Among fruits, apples are one of the richest in active compounds, mainly pectin (which acts as soluble fiber), amino acids, flavonoids (catechins and quercetin) and minerals (calcium, iron, magnesium, phosphorus and potassium), which provide apples with anti-inflammatory, antidiarrheal and anticancer properties, among others [2]. This, coupled with the high porosity of apples [3], has led to several studies in recent years aimed at obtaining apple snacks fortified with calcium [4], iron [5], antioxidants [6] and even probiotic microorganisms [7,8,9].

Probiotics are defined as ”live microorganisms which when administered in adequate amounts (10^8^−10^9^ CFU/day) confer a health benefit on the host” [10]. The major bacterial strains showing probiotic properties are lactic acid bacteria (LAB) from *Lactobacillus* and *Bifidobacterium* spp., which have been accorded the Generally Recognized as Safe (GRAS) status. The various health benefits documented for probiotics include suppression of obesity and type 2 diabetes through modulating glucose metabolism and prevention of liver disease, as well as suppression of hypercholesterolemia and cardio vascular disease through modulating total cholesterol, LDL-cholesterol and triglyceride metabolisms [11]. Other health disorders, such as allergic inflammation and atopic dermatitis; colon, bladder and cervical cancers; inflammatory bowel disease; diarrhea; lactose intolerance; renal diseases; hormonal immune response failure and poisoning caused by toxic compounds were also reported to improve with the ingestion of specific probiotic strains. Since viability of probiotic bacteria is required in order to exert health benefits in humans, ensuring that the different probiotic strains used in the formulation of foods are resistant to food manufacturing and storage conditions, as well as to the gastrointestinal digestion of the food, is a basic requirement when developing probiotic foods. Along with the processing temperature and the oxygen content in the product, the water activity (*a*_w_) is a key factor in maintaining probiotic viability in dry products [12,13,14,15,16,17]. As reported by several authors [12,13,15,16], viability of probiotic strains is lost rapidly with *a*_w_ > 0.25. They indicate optimal water activity values to be between 0.1 and 0.3, at which water might interact with functional groups and block reaction sites, thus avoiding interaction with oxygen and oxidation reactions that cause degradation of lipids and proteins of the probiotic cell. However, reaching such low water activity values is not an easy task in the case of fruits, due to their relatively high soluble solids content. In such cases, encapsulation [16,18] or the addition of various low-molecular weight additives (inulin, gum arabic, trehalose, sucrose, maltodextrin, etc.) [12,14,17,19,20,21] have been reported to significantly enhance the probiotic viability in intermediate water activity food products. Other techniques, such as the application of sub-lethal homogenization pressures (20–100 MPa) were also observed to increase the survival of probiotic strains in adverse conditions by inducing changes in the hydrophobicity of the cell membrane or in the availability of nutrients [22,23].

According to everything discussed above, this study aims to evaluate the effect that the drying technique and the water activity reached by the dried product has on the survival of *Lactobacillus salivarius* spp. *salivarius* and the stability of antioxidant properties of apple-based probiotic snacks during four-week storage under controlled conditions.

## 2. Materials and Methods

### 2.1. Raw Materials

Apples (*Malus domestica* cv. Granny Smith) used in the present study were purchased from a local market in Valencia, Spain. This variety was chosen for its high homogeneity and porosity, compared to other fruits. After being washed with tap water, apples were cut, following the longitudinal axis, into 5-mm-thick rings of 20 mm and 65 mm internal and external diameter, respectively.

*Lactobacillus salivarius* spp. *salivarius* supplied by the Spanish Culture Type Collection (strain CECT 4063, University of Valencia, Valencia, Spain) was selected as a probiotic microorganism in this study due to its potential effect against the infection caused by *Helicobacter pylori* and its ability to adaptat to the apple solid matrix [8]. Revival of the lyophilized strain was carried out in MRS Broth at 37 °C, following supplier’s recommendations. Then, the stock culture was plated on MRS Agar and incubated at 37 °C for 24 h. During the experiments, the plates were stored at 4 °C. When required, the white layer grown on the surface of each plate was collected in a test tube containing 9 mL of sterile MRS Broth and was further incubated at 37 °C for 24 h in order to obtain inoculums with a *Lactobacillus salivarius* spp. *salivarius* concentration on the order of (7.7 ± 0.9) × 10^8^ CFU/mL.

Commercial clementine juice (Hacendado brand, Valencia, Spain) was used in the impregnation liquid preparation. Following the procedure of Betoret et al. [8], 5 g/L of yeast extract and 9.8 g/L of sodium bicarbonate (both supplied by Scharlab, S.L., Barcelona, Spain) were added to the juice in order to ensure the appropriate growth of the microbial strain. One hundred milligrams per gram of food-grade trehalose from tapioca starch (TREHA TM, Cargill, SLU, Barcelona) were added in half of the preparations. After stirring at 200 rpm to get all the ingredients dissolved, the liquids were inoculated with 4 mL/L of MRS Broth containing the microorganism and incubated at 37 °C for 24 h to grow the acid lactic bacteria. Then, half of the liquid was homogenized at 100 MPa in a laboratory-scale high pressure homogenizer (Panda Plus 2000, GEA-Niro Soavi, Parma, Italy) and the other half was used directly in the vacuum impregnation step. Four different impregnation liquids were prepared in total: 0%_0MPa, 0%_100MPa, 10%_0MPa and 10%_100MPa, the percentage referring to the addition of trehalose and MPa to the homogenization pressure applied.

### 2.2. Snack Manufacturing Process and Storage

Vacuum impregnation (VI) was first carried out in a vacuum chamber (Heraeus Vacuun Oven, Thermo Fisher Scientific Inc.) connected to a vacuum pump (Gardner Denver Thomas GmbH Welch Vacuum, Fürstenfeldbruck, Germany) by applying a vacuum pressure of 50 mbar for 10 min to the apple rings immersed in the corresponding impregnation liquid (1:4 mass ratio) and then restoring the atmospheric pressure for another 10 min. Once impregnated, apple samples were stabilized by freeze-drying (FD) or convective drying at 40 °C until a water activity value of 0.45 or 0.35 (AD_0.45 and AD_0.35, respectively) was reached.

Freeze-drying was performed in two steps: freezing at −40 °C for 24 h in a CVN-40/105 Matek freezer and further sublimation at −45 °C and 0.1 mbar for 24 h in a 6–80 Telstar Lioalfa pilot plant scale freeze-dryer.

Convective drying was carried out in a CLW 750 TOP+ tray dryer (Pol-Eko-Aparatura SPJ, Vladislavia, Poland) with a cross flow of air at 2 m/s and 40 °C for about 24 or 48 h, depending on the water activity required at the end of the process.

Both freeze-dried and air-dried apple snacks were stored at room temperature in opaque and airtight bags for 4 weeks.

### 2.3. Analytical Determinations

#### 2.3.1. Moisture Content and Water Activity

Moisture content of apple samples was measured following the AOAC official method 934.06-1934 for dried fruits [24] and water activity was measured at 25 °C in a previously calibrated dew point hygrometer (Decagon Aqualab model CX-2, Pullman, WA, USA, with an accuracy of ±0.003).

#### 2.3.2. Antioxidant Properties

Antioxidant properties were determined for the extracts obtained by mixing a certain amount of apple (2 g vacuum-impregnated apple and 0.35 g for freeze-dried and air-dried apple) with 10 mL of an 80:20 (*v/v*) methanol of analytical high-performance liquid chromatography-grade (Merck KGaA and affiliates, Darmstadt, Germany) in water solution. After dispersing with a T 25 digital ULTRA-TURRAX^®^, the mixture was stirred in the dark at 200 rpm for 1 h and centrifuged at 10,000 rpm for 5 min at 4 °C in a Thermo Fisher Scientific Megafuge 16 centrifuge so as to obtain the methanolic extracts used in the antioxidant compounds quantification.

Total phenol content was determined following the method described by Singleton and Rossi [25] with some modifications. Briefly, 125 µL of the extract, 500 µL of bidistilled water and 125 µL of Folin–Ciocalteu phenol reagent (Merck KGaA and affiliates, Darmstadt, Germany) were mixed in a spectrophotometric cell. After incubation for 6 min in the dark, 1.25 mL of 7% Na_2_CO_3_ and 1 mL of bidistilled water were added. The mixture was kept at room temperature for 90 min in the dark and then the absorbance was measured at 760 nm on a Helios Zeta UV/Vis Thermo Scientific spectrophotometer (Waltham, MA, USA). Results were expressed as mg of gallic acid equivalents (purity ≥ 98%, Merck KGaA and affiliates, Darmstadt, Germany) per gram of dried sample by comparison with a standard calibration curve prepared in the range between 100 and 500 ppm (y = 0.0032x + 0.0305; R^2^ = 0.9963)

Total flavonoid content was determined according to the method described by Luximon-Rama et al. [26] with some modification. For this, 1.5 mL of the extract and 1.5 mL of a 2% (*w/v*) aluminum chloride (purity ≥ 98%) in methanol of analytical high-performance liquid chromatography-grade solution (both supplied by Merck KGaA and affiliates, Darmstadt, Germany) were mixed in a spectrophotometric cuvette. After 10 min reacting in the dark, the absorbance at 368 nm was measured on a Helios Zeta UV/Vis Thermo Scientific spectrophotometer. Results were expressed as mg of quercetin equivalents (purity ≥ 95%, Merck KGaA and affiliates, Darmstadt, Germany) per gram of dried sample by comparison with a standard calibration curve prepared in the range between 12.5 and 200 ppm (y = 0.0095x + 0.0799; R^2^ = 0.9977).

Finally, antioxidant activity of apple extracts was quantified according to their ability to scavenge the DPPH radical, as reported by Brand-Williams et al. [27]. Basically, 50 µL of the extract and 2950 µL of a 0.06 mM DPPH-methanol solution were mixed in a spectrophotometric cuvette and kept for 90 min in the dark before measuring the absorbance at 515 nm on a Helios Zeta UV/Vis Thermo Scientific spectrophotometer. Results were expressed as mg of Trolox equivalents (purity ≥ 97%, Merck KGaA and affiliates, Darmstadt, Germany) per gram of dried sample by comparison with a standard calibration curve prepared in the range between 100 and 300 ppm (y = −0.0014x + 0.4949; R^2^ = 0.9981).

#### 2.3.3. Color Measurements

Optical properties of apple samples were measured on a black background with a spectrocolorimeter (Minolta, CM-3600d), using D65 as illuminator and as 10° observer. Color measurements were given in CIE L*a*b* coordinates, where L* is the lightness that varies from black (0) to white (100), a* is the green (−) to red (+) color component and b* is the blue (−) to yellow (+) color component.

#### 2.3.4. Mechanical Properties

Mechanical properties of apple samples were evaluated by means of a puncture test in a TA-XT Plus texturometer (Stable Micro Systems, Godalming, United Kingdom). The test was carried out with a 2 mm diameter stainless steel cylindrical probe (Stable Micro Systems, Godalming, United Kingdom) that passed completely through the sample at a rate of 2 mm/s. From the force vs. distance curves provided by the equipment, the maximum force (*F*_max_, in N) and the distance travelled by the punch to reach the maximum force (*d*_max_, in mm) were obtained.

#### 2.3.5. Microbial Counts

*Lactobacillus salivarius* spp. *salivarius* content in both liquid and solid samples were estimated by serial dilution from 10^−1^ to 10^−6^ with peptone water, inoculation onto MRS Agar and incubation at 37 °C for 24 h. In the case of apple samples, the first dilution was obtained in a stomacher bag by mixing 5 g of sample with 45 mL of sterile peptone water (dilution 10^−1^) and blending at medium speed for 2 min.

#### 2.3.6. Statistical Analysis

Statistical analysis was carried out with the Statgraphics Centurion XVI program (Statgraphics Technologies, Inc., The Plains, VA, USA) by means of simple and multivariate analysis of variance (ANOVA) with a 95% confidence level.

## 3. Results and Discussion

### 3.1. Survival of Lactobacillus salivarius spp. Salivarius during the Snack Manufacturing Process

Table 1 shows the *Lactobacillus salivarius* spp. *salivarius* content of the different impregnation liquids, apple slices impregnated with each of them and subsequently freeze-dried or air-dried until reaching final water activity value. To better indicate the potential probiotic character of the different samples analyzed, concentrations were expressed on a wet basis instead of a dried basis. Together with experimentally obtained counts, theoretically calculated ones from the application of a mass balance in steady state, neglecting the generation term, were also calculated (Equations (1) to (3)).
(1)$${\left[x_{\mathrm{VIAPP}}^{\mathrm{mic}}\right]}_{\mathrm{THEO}}=\frac{X\cdot (1/\rho_{\mathrm{APP}})\cdot {\left[x_{\mathrm{VILIQ}}^{\mathrm{mic}}\right]}_{\mathrm{EXP}}}{1+X\cdot (\rho_{\mathrm{VILIQ}}/\rho_{\mathrm{APP}})}$$
(2)$${\left[x_{\mathrm{DEHAPP}}^{\mathrm{mic}}\right]}_{\mathrm{THEO}}=\frac{m_{\mathrm{VIAPP}}\cdot {\left[x_{\mathrm{VIAPP}}^{\mathrm{mic}}\right]}_{\mathrm{EXP}}}{m_{\mathrm{DEHAPP}}}$$
where ${\left[x_{\mathrm{VIAPP}}^{\mathrm{mic}}\right]}_{\mathrm{THEO}}$ and ${\left[x_{\mathrm{DEHAPP}}^{\mathrm{mic}}\right]}_{\mathrm{THEO}}$ are the theoretically calculated microbial contents of vacuum impregnated and dehydrated apples, respectively (CFU/g); ${\left[x_{\mathrm{VILIQ}}^{\mathrm{mic}}\right]}_{\mathrm{EXP}}$ and ${\left[x_{\mathrm{VIAPP}}^{\mathrm{mic}}\right]}_{\mathrm{EXP}}$ are the experimentally obtained microbial contents of impregnation liquids (CFU/mL) and vacuum impregnated samples (CFU/g), respectively; X is the impregnated volume fraction (reported by Fito et al. [3] to be 19 ± 1.5 m^3^ impregnation liquid/m^3^ fresh sample); $\rho_{\mathrm{VILIQ}}$ is the density of the impregnation liquid (reported by Betoret et al. [8] to be 1.085 ± 0.002 g/cm^3^); $\rho_{\mathrm{APP}}$ is the apparent density of fresh apple (reported by Fito et al. [3] to be 0.802 ± 0.010 g/cm^3^); $m_{\mathrm{VIAPP}}$ and $m_{\mathrm{DEHAPP}}$ are the total mass of vacuum impregnated and dehydrated apples, respectively (g); and $x_{\mathrm{VIAPP}}^{w}$ and $x_{\mathrm{DEHAPP}}^{w}$ are the experimentally measured moisture contents of vacuum-impregnated and dehydrated apples, respectively (g w/g).

Comparison between experimental and predicted values gives information about the influence that the different stages exert on the microbial counts, depending on the type of impregnation liquid used, which is expressed as a log reduction in Table 1.

Starting with liquid samples, a significant increase in the microbial content was observed by adding 10% (*w/w*) of trehalose to the juice before inoculation (10%_0MPa) or by homogenizing at 100 MPa the fermented juice (0%_100MPa). From this it follows that, as previously stated by other authors [28,29], trehalose serves as an important carbon and energy source, and as long as the osmotic pressure created in the growing media was not too high, it has the ability to enhance the growth status of several lactic acid bacteria strains. As regards the juice homogenization at sub-lethal pressures, it was previously reported to increase the microbial counts by reducing the size of the cloud particles and thus favoring the availability of nutrients [23]. In contrast, the combination of these two factors (10%_100MPa) did not significantly improve the viable counts.

As regards the *Lactobacillus salivarius* spp. *salivarius* content of vacuum impregnated samples, a reduction of around 0.8-log_10_ compared to that of vacuum impregnation liquids was observed. This result was expected because the external liquid is known to fill around 20% of the initial volume of apple slices during the vacuum impregnation step [3]. Furthermore, the similarity between predicted and experimentally obtained values for vacuum-impregnated apples (log reduction values close to zero) proves that vacuum impregnation is a useful technique for incorporating probiotics into a food solid matrix without negatively affecting their viability [8,9,30]. In accordance with the microbial counts reported for the impregnation liquids, apples impregnated with liquids 10%_0MPa and 0%_100MPa reached the highest content in *Lactobacillus salivarius* spp. *salivarius* after the vacuum impregnation step.

Finally, because of the water loss taking place during the dehydration step, the microbial content of dried apple samples was generally higher than that of the corresponding impregnated ones. However, based on the differences between predicted and experimental values, this increase was not as high as expected, thus showing the negative impact that drying has on the viability of the CECT 4063 strain. Among the drying techniques applied, freeze-drying had the least influence on the *Lactobacillus salivarius* spp. *salivarius* survival (0.6 ± 0.3-log_10_ reduction on average), and air drying at 40 °C until reaching a water activity around 0.35 had the greatest influence (2.2 ± 0.6-log_10_ reduction on average). This is mainly due to the lower exposure to oxygen and high temperature in the case of comparing freeze-drying with air drying, and to the shorter drying duration in the case of comparing air drying until reaching a water activity around 0.35 (48 h) with air drying until reaching a water activity around 0.45 (24 h). The loss of cell viability observed after freeze-drying might mainly be due to intracellular ice formation, which can damage the probiotic cell membrane [31]. The number of cells that survived the freeze-drying step ranged between 14.2% ± 0.7% and 61% ± 6% in apple samples impregnated with liquids 10%_100MPa and 10%_0MPa, respectively. In a similar study carried out with *Lactobacillus salivarius* spp. *salivarius* (UCC 500) solutions containing different substances as protective media [32], the survival rate immediately after freeze-drying was reported to increase from 4% in water without any protective agent to 34% in a suspension containing 4% of trehalose or even up to 83%–85% in a suspension containing 4% of sucrose, 4% of trehalose and 18% of skimmed milk. These results also suggest that, as previously indicated by Betoret et al. [33], the inclusion of only the probiotic into the apple’s porous structure by means of vacuum impregnation might confer it with protection against cell damage caused by freezing and subsequent sublimation of frozen water. This statement is not only supported by the higher viability obtained in the present study when the microorganism was subjected to freeze-drying as part of the apple porous structure (47% in apples impregnated with liquid 0%_0MPa) compared to that reported by Zayed and Roos [32] for the freeze-drying of the microorganism in just water, but also by the fact that a lower concentration of trehalose (≈ 2% *w/w* in the liquid phase of apples impregnated with liquid 10%_0MPa) resulted in a considerably higher survival of the microorganism to the freeze-drying step (78%) compared to that reported by Zayed and Roos [32] for the freeze-drying of the microorganism in a suspension containing 4% of trehalose. It is also possible that, since trehalose cannot be naturally synthetized by lactic acid bacteria [34], the beneficial effect of trehalose during freeze-drying was enhanced by its addition to the growth medium. Under less favorable drying conditions, as in the case of air-drying, the addition of 10% (*w/w*) of trehalose to the clementine juice before microbial inoculation resulted in less or no efficiency at all in preventing cell death, depending on the length of time exposure.

Regarding the homogenization of the fermented juice at 100 MPa, its effect on the CECT 4063 strain’s survival to drying was only significant when drying apple samples with air until reaching a water activity around 0.45 (AD_0.45 samples). This might be due to particle size reduction and related increased bioavailability of main compounds with antioxidant activity present in the juice [35]. As a result, probiotic cells would be more protected against oxidative stress caused by contact with the oxygen from the drying air. Likewise, this protective effect would not be as evident either in those samples not exposed to oxidation during drying, or in those samples exposed to excessive oxygen during drying. In such cases, the stress caused by the pressure gradient applied to the system seemed to be more harmful than beneficial for the probiotic survival to the drying process.

Besides the previous discussion, *Lactobacillus salivarius* spp. *salivarius* counts in dried apples were in all cases higher than 10^6^ CFU/g, which is the minimum concentration generally accepted for the probiotic benefits to be transferred to the consumer [36].

### 3.2. Antioxidant Properties Affected by the Snack Manufacturing Process

Antioxidant properties of apple slices subjected to vacuum impregnation with the different impregnation liquids and subsequently freeze-dried or dried with air at 40 °C until reaching a final water activity value are shown in Table 2.

As can be seen, the different liquids used in the vacuum impregnation step did not significantly affect either the total phenol and flavonoid content, or the overall antioxidant activity measured by the DPPH assay of vacuum-impregnated samples (VI APP). Antioxidant properties of apple slices generally improved after the dehydration step, but to a different extent, depending on the technique and the impregnation liquid previously used. Although slightly less pronounced, the increase in total phenol and flavonoid content and in the DPPH scavenging ability observed in freeze-dried samples could be explained in terms of a more efficient extraction of the antioxidant compounds. In fact, freeze-drying is often included in analytical procedures for use prior to the extraction of bioactive compounds from fruit matrices, while reducing the risk of isomerization and other undesirable reactions caused by their high enzymatic content [37]. Homogenization significantly increased total flavonoid release from the freeze-dried apple structure, but hardly affected that of total phenols, and even hindered that of other antioxidant compounds. It follows from this that particle size reduction and the release of antioxidant compounds from the complex structures in which they are retained due to high pressure homogenization [35] was particularly high in the case of total flavonoids. Meanwhile, the addition of trehalose to the juice formulation significantly increased total phenol and antioxidant release from the freeze-dried apple structure, but hindered that of total flavonoids. To explain this, reference can be made to the ability of trehalose to capture free radicals and prevent them from reacting with the reagents used in their analytical determination [22].

In the case of samples dried with air at 40 °C, moderate heating might have promoted the generation of new bioactive compounds or compounds with a greater antioxidant activity, as was reported for lycopene and β-carotene in cherry tomato halves undergoing osmotic dehydration at both 30 °C and 40 °C [38] or to polyphenols from granulated jaggery solutions subjected to heating below 100 °C for less than 20 min [39]. Although oxidation by long contact with the air stream was expected to cause antioxidant degradation, it might be negligible in this case compared to the aforementioned synthesis. Regarding the impregnation liquid used, neither the addition of trehalose to its composition nor the homogenization significantly improved the antiradical properties of air-dried apple snacks. However, the slight decrease observed in total phenol and flavonoid content when extending the drying time to reach a lower water activity was negligible (for total phenol content) or turned into a notable increase (for total flavonoid content) in those apples impregnated with liquids 0%_100MPa and 10%_0MPa. Impregnation of apple slices with liquids 0%_100MPa and 10%_0MPa also prevented Maillard reactions, which were reported to promote the formation of novel compounds having DPPH radical-scavenging activity [40]. On the contrary, combining the two factors in the same impregnation liquid seemed to favor both polyphenol degradation and Maillard reactions, as deduced from the significant lower values in total phenol and flavonoid content and the higher antioxidant activity obtained for AD_0.35 samples impregnated with the liquid 10%_100MPa.

### 3.3. Color Properties as Affected by the Snack Manufacturing Process

Instrumental color measurements revealed that the unit operations applied in the snack manufacturing process had a significant effect, whereas the type of impregnation liquid employed was irrelevant.

As shown in Figure 1, the vacuum impregnation of apple slices caused, regardless of the impregnation solution used, a significant decrease in the value of the L* coordinate. The replacement of the air occluded in the porous structure of apples by the external liquid implied it changing from whitish and opaque to bright and translucent. After subsequent dehydration, all three color coordinates rose markedly, most probably as a consequence of colored compound concentration and the oxidation reactions taking place during the dehydration of vacuum-impregnated samples. The increase in a* and b* coordinates means a change from greenish to a reddish tone. As expected, enzymatic and non-enzymatic browning reactions took place in a greater extent in air-dried than in freeze-dried samples, so the increase in the a* coordinate was significantly higher in the former ones. Furthermore, due to the longer exposure to the air required to achieve a lower water activity value (from 24 to 48 h), the rise in the a* coordinate was slightly higher in samples dehydrated to a lower water activity value (0.35). Water evaporation or sublimation taking place during the dehydration of vacuum impregnated samples explained the L* coordinate increase, with dehydrated apples becoming less translucent and more opaque than vacuum impregnated ones. The increase in the L* coordinate was greater in freeze-dried samples than in air-dried ones, thus suggesting that the water freezing and further sublimation favored a more porous structure than evaporation. On the contrary, air-drying might result in a more compact and less porous structure.

### 3.4. Mechanical Properties as Affected by the Snack Manufacturing Process

Force (N) vs. distance (mm) curves obtained after the puncture tests are shown in Figure 2. Since the multifactor ANOVA (*p*-value < 0.05) revealed no significant effect of the impregnation liquid on the maximum force and the distance to breakage values, only curves of those samples impregnated with liquid 0% _0MPa are presented as an example. As can be observed, values of both mechanical properties analyzed increased with the dehydration of vacuum impregnated apple slices, air-drying giving rise to significantly harder and more deformable snacks than freeze-drying. Furthermore, increasing the air-drying length in order to achieve lower water activity values (AD_0.35) caused a significant increase in the maximum force needed to break the sample, while slightly affecting the distance travelled by the punch. These results confirm that air-drying favors a rather rubbery and less crispy (crunchy/brittle) texture than freeze-drying. Indeed, the sublimation of water in freeze-dried samples resulted in the formation of a porous structure, whereas the evaporation of water during convective drying resulted in greater volume changes and structure collapse.

As regards the shape of the curves, that of vacuum impregnated samples was very similar to that of fresh and porous plant tissues, made up of tightly packed and turgid cells. After a slight deformation, the vacuum impregnated tissue underwent multiple fractures as the punch advanced through it. In the case of dehydrated samples, the force increased rapidly with the distance after a slight (for FD samples) or pronounced (for AD samples) initial deformation.

### 3.5. Survival of Lactobacillus salivarius spp. Salivarius during Snack Storage

Table 3 shows *Lactobacillus salivarius* spp. *salivarius* counts in dried apple samples after their storage for 7, 15 or 30 days under controlled conditions.

Although storage negatively affected microbial population in all cases, it was especially evident in those samples air dried at 40 °C until reaching a water activity of 0.45 (AD_0.45). Within a storage time of one month, the microbial viability was on average reduced 1.1 ± 0.5-log_10_ in FD samples, 0.9 ± 0.4-log_10_ in AD_0.35 samples and 4.3 ± 0.5-log_10_ in AD_0.45 samples. These results suggest that, beyond structural changes taking place during the dehydration step, the water activity of the final product exerts a decisive role in the survival of the microorganism during its subsequent storage. From our results, probiotic survival during storage was particularly affected when the food matrix had a water activity around 0.42 ± 0.02, which is well above the value of 0.25 reported to ensure the minimum loss of viability in dried products [13,41]. To explain the high survival of *Lactobacillus salivarius* spp. *salivarius* during storage in AD_0.35 samples, one could refer to the water activity value closer to the optimum or to the production of heat-shock proteins observed in many *Lactobacillus* strains submitted to sub-lethal heat and oxidative stress [42].

Regarding the effect of trehalose addition and/or the homogenization at 100 MPa on the CECT 4063 strain’s survival during storage, it was significantly affected by the water activity reached by the product at the end of the dehydration step. Survival of *Lactobacillus salivarius* spp. *salivarius* in both FD and AD_0.45 samples after 30 days of storage was not affected by the addition of trehalose and/or the homogenization. In contrast, cell survival in AD_0.35 samples improved significantly (*p*-value < 0.05) by including trehalose in the impregnation liquid composition, regardless of the application of a subsequent homogenization step (liquids 10%_0MPa and 10%_100MPa). Again, the production of heat-shock proteins by *Lactobacillus salivarius* spp. *salivarius* during long drying might be positively affected by sub-lethal osmotic stress induced when trehalose was added to the growing media. This hypothesis is reinforced by the synthesis of heat-sock proteins induced in *Lactobacillus delbrueckii* subsp. *bulgaricus* by acidic conditions [43] or in *Lactococcus lactis* by salt stress [44], which indicate that these proteins’ synthesis is a more general stress response, rather than just a thermal stress response. In all cases, the impregnation with the juice subjected to homogenization at 100 MPa after fermentation (liquid 0%_100MPa) resulted in the greatest loss of viability after 30 days of storage. This could be due not only to cell disruption caused by the application of a pressure gradient, but also to the increase in the availability of food bioactive compounds with antimicrobial properties [45].

### 3.6. Antioxidant Properties Change during Snack Storage

The net change in the antioxidant properties of the different probiotic snacks after 30 days of storage are shown in Figure 3.

The total phenol content increased 47% ± 2% in the case of freeze-dried samples (which had the highest counts at the end of the storage) but decreased in the case of air-dried ones. As previously reported for the content in *Lactobacillus salivarius* spp. *salivarius*, the loss of total phenols was significantly greater (*p*-value < 0.05) for those apple samples reaching a higher water activity at the end of the drying step (AD_0.45 samples). This direct relationship between the number of living cells and the content of total phenols was previously observed by Akman et al. [7] in samples impregnated with *Lactobacillus paracasei* and suggests that probiotic cells that were efficiently attached to the apple structure protected phenolic compounds from degradation caused by the attack of reactive oxygen species or by the polymerization of the monomeric phenolic compounds [46].

Total flavonoid content decreased in all cases, but to a different extent, depending on the composition of the vacuum impregnation liquid, the dehydration treatment applied and the interaction between both factors. Total flavonoid loss in samples impregnated with liquid 0%_0MPa was, regardless of the dehydration treatment applied, around 30% ± 3%. Adding 10% of trehalose to the impregnation liquid formulation or homogenizing it at 100 MPa significantly reduced the total flavonoid loss to 13% ± 3% in both FD and AD_0.45, but slightly increased it to 38% ± 6% in AD_0.35 samples. Combining both variables in liquid 10%_100MPa resulted in a flavonoid loss similar to that obtained for samples impregnated with liquid 0%_0MPa after one-month storage of AD_0.45 samples, but reduced it to 8.2% ± 1.5% in the case of FD and AD_0.35 samples.

Since the antioxidant activity is closely correlated with the presence of phenols and flavonoids [47], the DPPH radical scavenging activity also decreased in most of the cases after one month in storage. As stated previously for total phenols, the decrease in DPPH radical scavenging activity after storage was minimal for FD samples and maximal for AD_0.45 samples. Regarding the composition of the impregnation liquid, it was especially evident when storing FD apples, in which the juice homogenization at 100 MPa involved a significant increase in DPPH radical scavenging activity.

## 4. Conclusions

All the stabilization techniques applied in this study had a negative impact on *Lactobacillus salivarius* CECT 4063 counts, but generally improved the antioxidant properties of apple samples. Freeze-drying, especially after adding 100 mg/g of trehalose to the growth medium, proved to cause the lowest decrease in the microbial population. Meanwhile, drying with air at 40 °C up to a water activity of 0.45, especially when trehalose was not added to the growth medium and/or the vacuum impregnating liquid containing the microorganism was not homogenized, resulted in the greatest increase in both total phenol content (including flavonoids) and total ability to scavenge the DPPH radical.

Regarding the storage stability of *Lactobacillus salvarius* CECT 4063 in apple snacks, it was found to depend not only on water activity, but also on structural changes and/or heat-shock protein production induced by the different stabilization techniques applied. This is why the microbial count reduction within one month of storage was found to be higher in freeze-dried samples having the lowest water activity than in samples dried with air at 40 °C up to a water activity of 0.35. Moreover, cell survival in the latter case improved significantly (*p*-value < 0.05) by adding trehalose to the impregnation liquid composition.

As regards the antioxidant compounds, only the loss of total phenols within one month of storage was observed to increase significantly (*p*-value < 0.05) with the water activity of the apple snacks, whereas the loss of total phenols and total ability to scavenge the DPPH radical were more affected by the composition of the vacuum impregnation liquid, the dehydration treatment applied and the interaction between both factors. In any case, antioxidant compounds prove to have the maximum stability during the storage of freeze-dried samples.

## Figures and Tables

**Figure 1 microorganisms-08-01095-f001:**
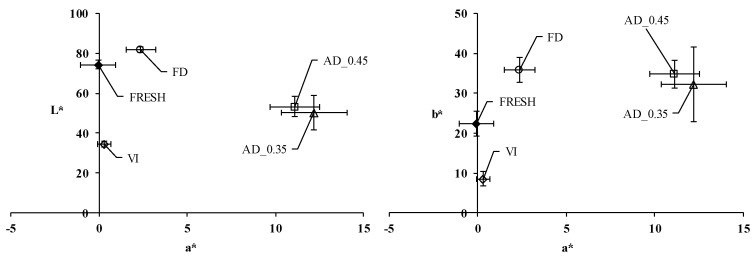
Colorimetric maps (L* vs. a* and b* vs. a*) of fresh apple (FRESH), vacuum-impregnated apple (VI), freeze-dried apple (FD) and air-dried apple at 40 °C until a final water activity of 0.45 (AD_0.45) or 0.35 (AD_0.35). Mean values of samples impregnated with the different impregnation liquids. Error bars represent the standard deviation of three replicates for each one of the different impregnation liquids.

**Figure 2 microorganisms-08-01095-f002:**
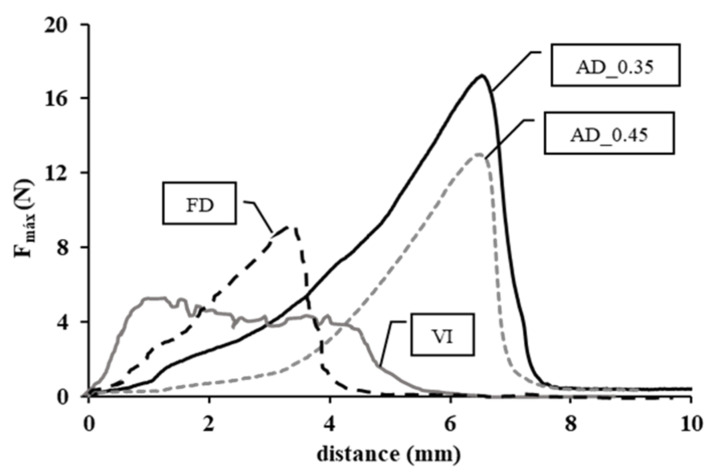
Typical force (in N) vs. distance (in mm) curves obtained after the puncture tests for apple samples impregnated with liquid 0% _0MPa (VI) and further freeze-dried apple (FD) or air-dried at 40 °C until a final water activity of 0.45 (AD_0.45) or 0.35 (AD_0.35).

**Figure 3 microorganisms-08-01095-f003:**
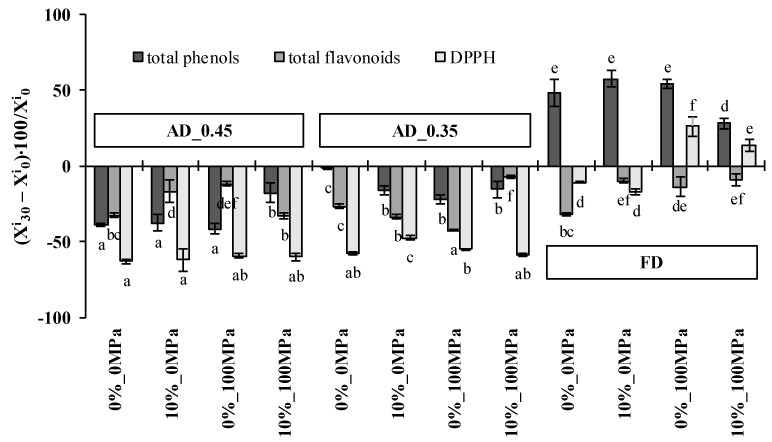
Antioxidant property change of apple snacks after 30 days of storage under controlled conditions. X^i^_0_ and X^i^_30_ respectively stand for the mass fraction in dry basis of component i at the beginning and at the end of the storage, VI APP stands for vacuum-impregnated apples, FD stands for freeze-dried apples, AD_0.45 stands for apples air-dried until reaching a *a*_w_ ≈ 0.45 and AD_0.35 stands for apples air-dried until reaching a *a*_w_ ≈ 0.35. Error bars represent the standard deviation of three replicates for each one of the different impregnation liquids. Different letters in the same series indicate statistically significant differences with a 95% confidence level (*p*-value < 0.05).

**Table 1 microorganisms-08-01095-t001:** Microbial counts in vacuum impregnating liquids (VI LIQ) and apple samples: comparison between predicted and experimental values. VI APP stands for vacuum impregnated apples, FD stands for freeze-dried apples, AD_0.45 stands for apples air-dried until reaching a water activity value *a*_w_ ≈ 0.45 and AD_0.35 stands for apples air-dried until reaching a *a*_w_ ≈ 0.35. Mean value of three replicates ± standard deviation. Different letters in the same column indicate statistically significant differences with a 95% confidence level (*p*-value < 0.05).

Sample	TRE%_HPH	*Log CFU/g*	
		*Experimental*	*Log Reduction*
**VI LIQ**	0%_0MPa 10%_0MPa 0%_100MPa 10%_100MPa	8.52 ± 0.02^fgh^ 9.1 ± 0.2^i^ 8.94 ± 0.12^hi^ 8.48 ± 0.07^fg^	-
**VI APP** *x*^w^ = 85.3 ± 1.2 g w/100 g	0%_0MPa 10%_0MPa 0%_100MPa 10%_100MPa	7.839 ± 0.009^de^ 8.20 ± 0.11^ef^ 8.1 ± 0.2^ef^ 7.7 ± 0.5^cd^	0.10 ± 0.02^j^ −0.09 ± 0.04^i^ −0.08 ± 0.04^i^ −0.05 ± 0.02^i^
**FD** *x*^w^ = 5.0 ± 1.0 g w/100 g	0%_0MPa 10%_0MPa 0%_100MPa 10%_100MPa	8.12 ± 0.10^ef^ 8.74 ± 0.06^ghi^ 8.39 ± 0.04^fg^ 7.68 ± 0.08^cd^	−0.53 ± 0.02^g^ −0.22 ± 0.04^h^ −0.64 ± 0.08^g^ −0.9 ± 0.3^f^
**AD_0.45** *x*^w^ = 11.4 ± 1.5 g w/100 g	0%_0MPa 10%_0MPa 0%_100MPa 10%_100MPa	7.46 ± 0.05^c^ 8.26 ± 0.03^ef^ 8.217 ± 0.004^ef^ 7.80 ± 0.04^cde^	−1.167 ± 0.009^e^ −1.46 ± 0.03^cd^ −1.60 ± 0.07^c^ −1.25 ± 0.02^de^
**AD_0.35** *x*^w^ = 9.2 ± 0.8 g w/100 g	0%_0MPa 10%_0MPa 0%_100MPa 10%_100MPa	7.5 ± 0.3^cd^ 6.25 ± 0.08^a^ 7.02 ± 0.03^b^ 6.10 ± 0.02^a^	−1.337 ± 0.009^de^ −2.68 ± 0.03^a^ −2.10 ± 0.07^b^ −2.63 ± 0.02^a^

**Table 2 microorganisms-08-01095-t002:** Antioxidant properties of apple samples after each stage of the snack manufacturing process. VI APP stands for vacuum impregnated apples, FD stands for freeze-dried apples, AD_0.45 stands for apples air-dried until reaching a *a*_w_ ≈ 0.45 and AD_0.35 stands for apples air-dried until reaching a *a*_w_ ≈ 0.35. Mean value of three replicates ± standard deviation. Different letters in the same column indicate statistically significant differences with a 95% confidence level (*p*-value < 0.05).

Sample	TRE%_HPH	Total Phenols (mg GAE/g dw)	Total Flavonoids (mg QE/g dw)	Antioxidant Activity (mg TE/g dw)
**VI APP** *a*_w_ = 0.983 ± 0.002	0%_0MPa 10%_0MPa 0%_100MPa 10%_100MPa	5.4 ± 0.7^a^ 5.7 ± 1.2^abc^ 5.7 ± 0.5^ab^ 5.3 ± 1.5^a^	1.24 ± 0.15^abc^ 1.07 ± 0.06^a^ 1.3 ± 0.3^abc^ 1.2 ± 0.4^ab^	7.0 ± 0.9^bcdef^ 7 ± 3^bcde^ 6.0 ± 0.9^bc^ 6.4 ± 1.8^bcd^
**FD** *a*_w_ = 0.25 ± 0.02	0%_0MPa 10%_0MPa 0%_100MPa 10%_100MPa	5.4 ± 0.3^ab^ 8.39 ± 0.03^f^ 6.3 ± 0.3^abcd^ 6.6 ± 0.5^abcde^	1.584 ± 0.013^cd^ 1.187 ± 0.013^abc^ 1.76 ± 0.12^de^ 1.74 ± 0.03^de^	8.64 ± 0.07^efghi^ 10.3 ± 0.4^i^ 4.9 ± 0.3^ab^ 3.6 ± 0.5^a^
**AD_0.45** *a*_w_ = 0.42 ± 0.02	0%_0MPa 10%_0MPa 0%_100MPa 10%_100MPa	10.6 ± 0.3^g^ 6.7 ± 0.2^bcde^ 7.6 ± 0.2^def^ 7.1 ± 0.5^cdef^	2.47 ± 0.09^g^ 1.51 ± 0.06^bcd^ 2.13 ± 0.09^efg^ 1.53 ± 0.02^cd^	10.2 ± 0.4^i^ 9 ± 2^ghi^ 6.9 ± 0.2^bcdefg^ 8.5 ± 0.4^defghi^
**AD_0.35** *a*_w_ = 0.36 ± 0.03	0%_0MPa 10%_0MPa 0%_100MPa 10%_100MPa	8.1 ± 0. 2^ef^ 6.6 ± 0.5^abcde^ 7.8 ± 0.4^def^ 5.867 ± 0.014^abc^	1.990 ± 0.006^ef^ 1.876 ± 0.012^de^ 2.36 ± 0.07^fg^ 1.044 ± 0.005^a^	9.2 ± 0.3^fghi^ 7.68 ± 0.15^cdefgh^ 6.20 ± 0.07^bcde^ 9.5 ± 0.3^hi^

**Table 3 microorganisms-08-01095-t003:** Microbial counts in apple snacks during their storage for 30 days under controlled conditions. FD stands for freeze-dried apples, AD_0.45 stands for apples air-dried until reaching a *a*_w_ ≈ 0.45 and AD_0.35 stands for apples air-dried until reaching a *a*_w_ ≈ 0.35. Mean value of three replicates ± standard deviation. Different letters in the same column indicate statistically significant differences with a 95% confidence level (*p*-value < 0.05).

Sample	TRE%_HPH	Storage Time			Log Reduction
		7 Days	15 Days	30 Days	
**FD** *a*_w_ = 0.25 ± 0.02	0%_0MPa 10%_0MPa 0%_100MPa 10%_100MPa	7.80 ± 0.05^c^ 8.23 ± 0.06^d^ 7.75 ± 0.04^c^ 7.84 ± 0.02^c^	7.66 ± 0.11^fg^ 7.84 ± 0.04^g^ 7.46 ± 0.12^f^ 7.76 ± 0.06^g^	7.35 ± 0.06^h^ 7.10 ± 0.10^gh^ 6.73 ± 0.02^fg^ 6.71 ± 0.11^g^	−0.76 ± 0.06^g^ −1.61 ± 0.10^de^ −1.66 ± 0.02^d^ −0.65 ± 0.11^gh^
**AD_0.45** *a*_w_ = 0.42 ± 0.02	0%_0MPa 10%_0MPa 0%_100MPa 10%_100MPa	5.24 ± 0.09^a^ 6.05 ± 0.04^b^ 6.11 ± 0.02^b^ 5.2 ± 0.4^a^	5.1 ± 0.7^b^ 5.7 ± 0.2^d^ 5.4 ± 0.4^cd^ 4.2 ± 0.4^a^	4.1 ± 0.4^bc^ 4.1 ± 0.4^c^ 3.27 ± 0.13^a^ 3.46 ± 0.09^ab^	−3.6 ± 0.2^c^ −4.2 ± 0.4^b^ −4.95 ± 0.13^a^ −4.34 ± 0.09^b^
**AD_0.35** *a*_w_ = 0.36 ± 0.03	0%_0MPa 10%_0MPa 0%_100MPa 10%_100MPa	6.32 ± 0.07^b^ 5.72 ± 0.02^a^ 6.00 ± 0.08^a^ 5.83 ± 0.05^b^	6.261 ± 0.012^e^ 5.64 ± 0.10^d^ 5.889 ± 0.012^e^ 5.58 ± 0.05^c^	6.09 ± 0.14^ef^ 5.24 ± 0.05^d^ 5.22 ± 0.07^de^ 5.18 ± 0.10^d^	−1.21 ± 0.07^f^ −0.53 ± 0.02^gh^ −1.4 ± 0.4^ef^ −0.52 ± 0.05^h^

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
