# Peer review of "Survival of Lactobacillus salivarius CECT 4063 and Stability of Antioxidant Compounds in Dried Apple Snacks as Affected by the Water Activity, the Addition of Trehalose and High Pressure Homogenization"

_microorganisms, 2020, doi:10.3390/microorganisms8081095_

Round 1
Reviewer 1 Report
This is a very interesting paper on the "Survival of Lactobacillus salivarius CECT 4063 and the stability of antioxidant compounds in dried apple snacks as affected by the water activity, the addition of trehalose and high pressure homogenization.
However in a literature search regarding trehalose and probiotics I came across the article https://doi.org/10.1051/dst:2007003. I am not among the authors nor do I have any connection with them. However this paper is not cited or discussed. Hence, a proper check of previous literature is recommended.
A section regarding the chemicals (Folin, AlCl3, gallic, quercetin, MeOH, etc) used in this study with their origin (purchase) should be included in the materials and methods section.
Since determination was based on Standrad curves, their equations and the corresponding R value should be reported.
In your std curves you use ppm, while results are expressed as mg/g dry weight. The original values acquired by std curves and dry weight of sample as well as total weigt of sample used should be reported. Please clarify.
P3 L 124 Please provide centrifuge temperature and type
L 160 Serial dilution of solid samples. Please clarify
Since you present results and discussion together, I think that a closing paragraph with the main conclusions of the study would benefit your work.
Author Response
This is a very interesting paper on the "Survival of Lactobacillus salivarius CECT 4063 and the stability of antioxidant compounds in dried apple snacks as affected by the water activity, the addition of trehalose and high pressure homogenization.
The authors appreciate the sincere opinion of reviewer 1 and greatly thank him the effort made to improve the manuscript.
However, in a literature search regarding trehalose and probiotics I came across the article https://doi.org/10.1051/dst:2007003. I am not among the authors nor do I have any connection with them. However, this paper is not cited or discussed. Hence, a proper check of previous literature is recommended.
Following reviewer 1 recommendations, the bibliographic search has been extended and new references evaluating the effect of water activity and low-molecular weight additives on probiotic survival to both processing and storage have been added to the text.
Miao, S., Mills, S., Stanton, C., Fitzgerald, G.F. Roos, Y., & Ross, R.P. (2008). Effect of disaccharides on survival during storage of freeze dried probiotics. Dairy Science & Technology 88(1), 19-30. DOI: 10.1051/dst:2007003
Bagad, M., Pande, R., Dubey, V., & Ghosh, A.R. (2017). Survivability of freeze-dried probiotic Pediococcus pentosaceus strains GS4, GS17 and Lactobacillus gasseri (ATCC 19992) during storage with commonly used pharmaceutical excipients within a period of 120 days. Asian Pacific Journal of Tropical Biomedicine 7(10), 921-929. DOI: 10.1016/j.apjtb.2017.09.005
Golowczyc, M.A., Gerez, C.L., Silva, J., Abraham, A.G., De Antoni, G.L., & Paula Teixeira, P. (2011). Survival of spray-dried Lactobacillus kefir is affected by different protectants and storage conditions. Biotechnology Letters 33, 681-686. DOI: 10.1007/s10529-010-0491-6
Lapsiri, W., Bhandari, B., & Wanchaitanawong, P. (2012) Viability of Lactobacillus plantarum TISTR 2075 in Different Protectants during Spray Drying and Storage. Drying Technology 30(13), 1407-1412. DOI: 10.1080/07373937.2012.684226
A section regarding the chemicals (Folin, AlCl3, gallic, quercetin, MeOH, etc) used in this study with their origin (purchase) should be included in the materials and methods section.
Instead of mentioning all the reagents and solvents in a single section within the materials and methods chapter, authors have decided to indicate one by one purity and supplier as they appear in the text (lines 123 & 124, 130, 134 & 135, 139 & 140, 143, 150 & 151).
Since determination was based on Standard curves, their equations and the corresponding R value should be reported.
Equations of the standard curves and the corresponding coefficient of determination (R2) have been reported following the reviewer 1 recommendation (lines 136, 145, 152 & 153).
In your std curves you use ppm, while results are expressed as mg/g dry weight. The original values acquired by std curves and dry weight of sample as well as total weight of sample used should be reported. Please clarify.
The total weight of sample used in analytical determination of antioxidant properties is already provided in the text, as well as the amount of solvent employed for the extraction of the antioxidant compounds (lines 122 & 123). Moisture content of apple samples both VI, FD, AD_0.45 or AD_0.35 are also reported in table 1, therefore the amount of dry solid in which the antioxidant compounds are retained can be easily estimated. Having added now the equations of the three standard curves, authors do not consider it also necessary to report the absorbance values read at the different wavelengths since they could be mathematically deduced from the information provided.
P3 L 124 Please provide centrifuge temperature and type
Centrifugation temperature and centrifuge type have been already added to the materials and methods section (line 126).
L 160 Serial dilution of solid samples. Please clarify.
Information about the dilution range has been added to the text (line 168). Clarification on how to prepare dilution 10-1 from solid apple samples has been also provided (line 170).
Since you present results and discussion together, I think that a closing paragraph with the main conclusions of the study would benefit your work.
A conclusions section has been added to the manuscript (lines 433-454).
Reviewer 2 Report
The manuscript is original research about the analysis of various treatment methods on survival of Lactobacillus salivarius and stability of antioxidant compounds in dried apple snacks. The topic is noteworthy and very interesting for readers of Microorganisms. The work is very comprehensive, well written and supported by good experimental design. The objective of the paper is clearly defined. The experimental apparatus is standard and appropriate for the study. The methods are well described and provide necessary information to reproduce the experiments. Most results are clearly explained and presented in an appropriate format.
I found a few points that should be supplemented or improved:
- What statistical test was used to analyze the data?
- In Tables 1 and 3 all letters indicating the significance of differences between means should be marked.
- Figure 3. How to explain the increase in total phenolic content and the simultaneous decrease in radical scavenging activity of apple snack after freeze drying?
- There is a lack of conclusions.
The manuscript can be published after minor revision.
Author Response
The manuscript is original research about the analysis of various treatment methods on survival of Lactobacillus salivarius and stability of antioxidant compounds in dried apple snacks. The topic is noteworthy and very interesting for readers of Microorganisms. The work is very comprehensive, well written and supported by good experimental design. The objective of the paper is clearly defined. The experimental apparatus is standard and appropriate for the study. The methods are well described and provide necessary information to reproduce the experiments. Most results are clearly explained and presented in an appropriate format.
Thank you very much. Authors really appreciate reviewer 2 opinion.
I found a few points that should be supplemented or improved:
What statistical test was used to analyze the data?
Simple and multivariate analysis of variance (ANOVA) with a 95% of confidence level was applied to analyse the effect of the different processing variables considered (lines 172-174).
In Tables 1 and 3 all letters indicating the significance of differences between means should be marked.
All letters indicating the significance of differences between means are already added in tables 1 and 3.
Figure 3. How to explain the increase in total phenolic content and the simultaneous decrease in radical scavenging activity of apple snack after freeze drying?
Total phenols are not the only antioxidants present in fresh and processed apples that are able to scavenge free radicals but there are others, mainly vitamin C (Eberhardt et al., 2000). Moreover, the antioxidant activity of phenolics is not only due to their content but mainly to their chemical structure, which can vary along processing and storage (Bahukhandi et al., 2018). According to this, the increase in total phenolic content and the simultaneous decrease in radical scavenging activity of freeze-dried apples along their storage could be explained in terms of vitamin C degradation and/or polyphenols generation having a lower antioxidant activity.
Eberhardt, M.V., Lee, C.Y., & Rui Hai Liu, R.H. (2000). Antioxidant activity of fresh apples. Nature 405, 903-904.
Bahukhandi, A., Dhyani, P., Bhatt, I.D., & Rawal, R.S. (2018). Variation in Polyphenolics and Antioxidant Activity of Traditional Apple Cultivars from West Himalaya, Uttarakhand. Horticultural Plant Journal 4(4), 151-157.
There is a lack of conclusions.
A conclusions section has been added to the manuscript (lines 433-454).